# Effects of Barefoot Walking in Urban Forests on CRP, IFNγ, and Serotonin Levels

**DOI:** 10.3390/healthcare12232372

**Published:** 2024-11-26

**Authors:** Jae Sun Kim, Mi Mi Lee, Dong Soo Kim, Chang Seob Shin

**Affiliations:** 1Department of Forest Therapy, Graduate School of Chungbuk National University, Cheongju 28644, Republic of Korea; 2Ecological Meditation Community, Goyang 10450, Republic of Korea; yulim741@naver.com; 3Department of Natural Science, Korean Air Force Academy, Cheongju 28100, Republic of Korea; dongsookim04@gmail.com; 4Department of Forest, Chungbuk National University, Cheongju 28644, Republic of Korea

**Keywords:** barefoot walking exercise, CRP, NK-cell, interferon gamma, IFNγ, serotonin

## Abstract

Background/Objectives: This study investigated the effects of barefoot walking on an urban forest path on participants’ C-reactive protein (CRP), interferon gamma (IFNγ), and serotonin levels, which are associated with feelings of well-being. Methods: Participants in the walking program included 62 consenting adults, divided into a sneaker-wearing and barefoot group (each *N* = 31). The program comprised 20 sessions, each consisting of a 90 min 4.4 km walk at a 50–70% target heart rate, held four times per week for 5 weeks. Physiological measurements were taken from pre-program saliva samples and after 10 and 20 sessions, respectively. Results: The sneaker-wearing group showed a consistent CRP increase, with levels reaching statistical significance after 20 sessions. In the barefoot group, CRP increased up to Session 10 but was lower than at the end of Session 20. The analysis of those with an initial CRP above 100 pg/mL showed that CRP was statistically significantly lower in the barefoot (*N* = 14) than in the sneaker-wearing (*t* = −0.048, *p* = 0.963) group after 20 sessions compared to before the program (*t* = 3.027, *p* = 0.010). IFNγ increased in both groups up to Session 10 but showed minimal change between Sessions 10 and 20. Serotonin was statistically significantly higher after 20 sessions than before the program in the barefoot group (*t* = −2.081, *p* = 0.046). Conclusions: Barefoot walking on forest trails is effective for people with above-normal inflammation, increasing serotonin levels.

## 1. Introduction

In modern society, ecosystems have been destroyed by indiscriminate resource development caused by rapid industrialization since the nineteenth century, which has led to more exposure to a variety of unpredictable environmental diseases, infectious bacteria, and viruses. In response, the development of new vaccines and disease-preventative actions has continued in a seemingly vicious cycle.

This is a result of the ongoing disconnect between humans and the natural environment. Kaplan and Kaplan’s attention restoration theory (ART) suggests that the natural environment is a place to escape from the complexity and busyness of modern life and represents a space for attention restoration [1]. Ulrich’s psycho-evolution theory (PET) suggests that humans have long evolved in natural environments and have adapted to respond positively physiologically and psychologically to natural environments rather than urban environments [2]. In a cohort study of patients who underwent gallbladder removal surgery at a Pan-Sylvania hospital between 1972 and 1981, those who had a window view of the forest recovered much faster than those who did not, suggesting that the natural environment itself plays an important role in reducing human stress [3]. In addition, the theory of forest stimuli suggests that stimuli in forests bring about positive psychological changes in people by triggering coping behaviors that allow them to respond wisely and deal with problems in an appropriate way [4]. In his savanna theory, Orians noted that humans have a preference for certain trees, shrubs, or plants in savanna-like environments [5,6].

In Korea, the “forest healing” concept was introduced to realize practical health promotion in forests in 2012. It is defined as “an activity to improve human immunity and health using various elements of nature, such as fragrance and scenery” [7].

The healing elements of forests include scenery, phytoncide, anions, oxygen, sound, and sunlight. Kim et al. [8] classified forest healing methods as plant, climate, and water therapies, all of which utilize forest resources, as well as diet, exercise, and psychotherapy, which can be practiced in forests. Among these, exercise therapy has three subcategories: experience type, barefoot walking, and forest gymnastics.

Forest walking is a simple daily activity that utilizes inclined rocks and branches that naturally exist in the forest terrain. It requires no special equipment or space and is among the most common, simple, economical, and beneficial exercises.

In a study related to forest walking, Kang’s [9] research subjects performed forest trekking biweekly for 10 weeks and observed significant increases in dynamic equilibrium and NK cell activation. Shin et al. [10] found that 12-week forest slope movement and phytoncide affected the brain-derived neurotrophic factor that accelerates the growth of brain cells for cognitive function. Shin [11] found that a 15 min walk in a forest environment positively improved heart rate variability, mood, and emotional state in patients with heart disease. Lee [12] found that flexibility, agility, equilibrium, cardiopulmonary endurance, self-efficacy, and stress improved after 12 weeks of forest-walking exercise. Furthermore, Lee [13] observed effective change in the T and NK cells involved in immune function after 12 weeks of forest-walking exercise.

Exercising in a forest environment is known to be more beneficial to immunity and health than walking on flat land owing to forests’ healing factors and the forest terrain’s exercise effect.

The recently highlighted effects of barefoot walking can be divided into two categories: the foot reflexology effect, which occurs as the ground stimulates the soles of the feet during barefoot walking, and the earthing effect, which arises when the soles of the feet make contact with the ground.

Regarding the foot reflexology effect, So [14] found improved function in all body parts after stimulating the stimulation points of the reflex zone distributed over the soles of the feet. Research has also reported that stimulating the soles of the feet improves heart function [15], reduces blood triglyceride, which hinders blood circulation [16], and activates blood flow throughout the body by expanding blood vessels [17]. Ober et al. [18] found that contact between the human body and the Earth (ground) restores natural electrical stability and rhythm, allowing the body’s systems, including the circulatory, respiratory, digestive, and immune systems, to function normally. According to Oschman et al. [19], earthing improves sleep and contributes to cortisol rhythm normalization during the day and night, pain and stress reduction, the conversion of the autonomic nervous system from sympathetic to parasympathetic, increased heart rate variability, faster wound healing, and blood viscosity.

Related domestic studies on barefoot walking include research that compared the effects of barefoot walking on physical fitness, blood pressure, hyperglycemia, and obesity, all of which are metabolic syndrome indicators [20]. Other notable studies include those by Kim [21] and Choi [22], as well as research on changes in back pain [23], brain activity, and psychological changes [24]. Further studies include those of Lee [25], Han [26], and Jung [27], as well as research on improved immunity [28] and stress and blood vessel health [29].

Most of the previous barefoot walking case studies have been conducted in schoolyards or beaches. Research on barefoot walking in forests has been limited to psychological changes in metabolic diseases. Therefore, the effect of barefoot walking in a forest environment on the body’s immune system is not yet understood. Exposure to barefoot forests has also been linked to health benefits, including reduced cortisol levels through stress reduction, increased alpha waves at rest, NK cells to boost immunity, and superoxide dismutase to help fight aging.

In general, the C-reactive protein (CRP) is a substance that causes acute inflammation or tissue damage in the body; the normal salivary level is 61.29 ± 7.38 pg/mL. Meanwhile, the normal level of interferon gamma (IFNγ) related to the immune system is 5.0–45.0 pg/mL; it contributes to the anti-cancer activity of NK cells among immune cells. In addition, serotonin, a neurotransmitter, is considered normal within the range of 0.005–0.045 ng/mL.

This study aimed to verify whether physiological changes occurred when barefoot walking was performed in an urban forest environment. To this end, adults participated in a walking program, and changes in the levels of the following factors were analyzed: CRP, an indicator of the inflammation level and the possibility of diseases; IFNγ, an NK cell activity indicator; and serotonin, which is known as the happiness hormone.

The research hypothesis of this study is that barefoot walking in urban forests results in statistically significant changes in CRP, IFNγ, and serotonin levels.

## 2. Materials and Methods

### 2.1. Participants

G-POWER 3.1.9.4 was used to calculate the appropriate number of participants to achieve statistical significance [30]. A 52-person recommended sample size was obtained by calculation with an effect size of 0.8, a significant probability of 0.05, and a statistical power of 0.8. The experiment dropout rate was set to 20% following a previous study [31]. The final sample size, in consideration of the dropout rate, was 62 people. Participants were recruited via distributed leaflets and local networks. The experiment involved 62 participants, who were randomized between sneaker-wearing and barefoot-walking groups after two-stage stratified sampling that first took into account age and sex. Participants were healthy adults with no walking problems or barefoot walking experience. They were selected under the condition that they would refrain from other exercise during the experiment period.

Before conducting the research, the research purpose and contents were sufficiently explained to the participants, and their written consent was obtained. For safety, a tetanus vaccination was given to those participants who wanted it before the commencement of the experiment. Table 1 shows the research participants’ details.

### 2.2. Site

The research was conducted in the Jeongbal Mountain (Jeongbalsan) urban forest. The mountain has an altitude of 87 m. The site is in Ilsandong-gu, Goyang-si, Gyeonggi-do, Korea. This urban park is adjacent to residential areas. It falls within Korea’s central temperate zone (37°39′42″ N, 126°46′42″ E) and covers 644,649 m^2^. Green areas account for over 91% of the park’s total area, and the main tree species are pine trees, Korean pine trees, and metasequoia. The park is a recreational and exercise space used to promote urban citizens’ health. The walkable distance at Jeongbalsan Dulle-gil is 4.4 km, with an average slope of 7.5°. The trails that run north and west comprise sandy loam, and the trails that run south and east consist of fine red clay. The “rest road” is safe for barefoot walking and was created under the Goyang City “earthing road” project in 2021. Figure 1 provides further information about the site.

### 2.3. Methodology

This study sought to detect physiological changes in the subjects by varying the form of their walking exercise on urban forest trails. The walking program comprised 20 sessions over 5 weeks at a frequency of four sessions per week. Each session included a 5 min warm-up exercise, the 90 min main exercise, which involved walking 4.4 km, and a 5 min cool-down exercise, for a session total of 100 min. Before the commencement of the experiment/program, a saliva sample was collected from each participant using Salivette. Additional saliva samples were collected after the participants had completed 10 and 20 sessions, respectively; that is, they were collected at the midpoint and end of the experiment/program. The samples were frozen and analyzed by an analysis agency. The experiment period spanned 21 July 2023 to 25 September 2023, including an additional recruitment period for subjects who dropped out of the program (4 people). Figure 2 illustrates the experimental procedure. See Appendix A Figure A1 and Figure A2 for the detailed experimental procedure.

### 2.4. Walking Program

#### 2.4.1. Program Configuration

The walking program was designed for this research with reference to the American College of Sports Medicine’s exercise guidelines [32], the details of which are visualized in Table 2. The workout intensity was set at a 50–70% target heart rate (THR), and the rate of perceived exertion was set to 10–13. Figure 3 shows participants undertaking the program.

#### 2.4.2. Exercise Intensity Measurement

The sneaker-wearing and barefoot groups performed the experiment in the same environment and under identical conditions. The smartphone walking management application Polar Beat was operationalized before the participants started walking. Upon completion of the exercise, the application stored each participant’s actual exercise time, distance, workout intensity, average heart rate, and maximum heart rate. Additionally, a Bryton heart rate monitor was attached to each person’s chest once per week to adjust the workout intensity to ensure that the actual heart rate remained within 50–70% of each individual’s THR. Analyzing the stored exercise intensity data protected the experimental data’s objectivity and reliability. Figure 4 shows the method used to measure walking exercise intensity.

### 2.5. Analyses

#### 2.5.1. Saliva Analysis

Saliva samples were collected as described in Section 2.3. Participants were disallowed from consuming food or water for 1 h before the samples were collected. Tooth brushing and gargling were also prohibited because bleeding in the mouth could contaminate the samples. Immediately before the samples were collected, participants rinsed their mouths with water, and a cotton swab was placed under each person’s tongue. Swabs that had absorbed a sufficient amount of saliva were frozen and analyzed by an agency that measured the contents using the enzyme-linked immunosorbent assay (ELISA). Detailed experiments were performed in accordance with the procedure the ELISA kit vendor specified. An ELISA reader (Biotek Instruments, Winooski, VT, USA) was used during the analysis. Table 3 presents the analyzed items.

#### 2.5.2. Data Analysis

The sneaker-wearing and barefoot groups were compared to investigate the physiological changes in the research subjects through participation in the forest-walking exercise program. SPSS 27.0 was used for data analysis. Frequency analysis, chi-squared (χ2), and independent sample *t*-testing were utilized to examine the participants’ general characteristics and homogeneity. A paired sample *t*-test was conducted to compare the two groups, and the statistical significance level was set to *p* < 0.05.

## 3. Results

### 3.1. General Characteristics

The research subjects’ general characteristics were analyzed via a preliminary survey, which was distributed to all 62 participants. The survey items covered age, sex, alcohol consumption and smoking habits, usual exercise status, exercise frequency, exercise duration, subjective health status, eating habits, and food preferences. Frequency analysis, chi-squared (χ2), and independent sample t-testing were conducted for statistical analysis. No between-group differences, in general, were found for the participants’ characteristics (Table 4).

These results confirmed that the participants would have no physical problems performing the walking experiment, as they entered the program with a stable lifestyle and regular exercise.

### 3.2. Preliminary Homogeneity Test

#### 3.2.1. Physical Characteristics

Before the experiment, an independent sample *t*-test was conducted to ascertain the homogeneity of the sneaker-wearing and barefoot groups’ physical characteristics. Table 5 shows the results.

The statistical significance level for all variables is *p* > 0.05, indicating that the two groups are homogeneous with no differences in physical characteristics.

#### 3.2.2. Physiological Characteristics

Before the experiment, an independent sample *t*-test was conducted to ascertain the homogeneity of the sneaker-wearing and barefoot groups’ physiological characteristics. Table 6 shows the results.

The level of statistical significance for all variables was *p* > 0.05, indicating that the two groups were homogeneous with no differences in their physiologic characteristics.

#### 3.2.3. Normality Tests

Normality tests for skewness and kurtosis, as well as the Shapiro–Wilk test, were conducted for 31 subjects, excluding two missing values per group. Table 7 shows the results.

To test for normality, we applied the commercial logarithm to the mean and examined the skewness and kurtosis. As Table 7 shows, both populations satisfied a skewness < 2 and kurtosis < 4, and thus, we met the conditions to run a paired *t*-test to compare the arithmetic means between the two populations.

### 3.3. Physiological Change Measurement Results

#### 3.3.1. CRP

To investigate the change in the level of the inflammation indicator CRP in the subjects’ saliva, their saliva samples were, as previously mentioned, analyzed three times: before the commencement of the walking program, at the program’s midpoint (after completing 10 sessions), and at the end of the program (after completing 20 sessions). Table 8 shows the results of a paired sample *t*-test conducted to identify changes in the two groups.

The results show that the sneaker-wearing walking group showed a statistically significant increase in inflammation, while the barefoot-walking group showed an initial increase and then a decrease after 10 sessions (See Figure 5).

#### 3.3.2. CRP-100

Table 9 shows the results of analyzing CRP changes after selecting, from among all 62 participants, only those who exhibited a saliva CRP inflammation level of 100 pg/mL or higher before the experiment. The targets included 14 people from the barefoot group and 9 from the sneaker-wearing group. A paired sample *t*-test was conducted to examine the changes.

A comparison of the two groups showed a statistically significant decrease in the barefoot group, suggesting that barefoot walking should be continued for at least 10 weeks to achieve a reduction in inflammation. (See Figure 6).

#### 3.3.3. IFNγ

Table 10 shows the results of analyzing the changes in the salivary level of IFNγ, which is an anti-cancer marker.

The results show that the levels of IFNγ changed from the beginning of the experiment, indicating that 10 sessions were enough to induce an effect, while after 10 sessions, there was no change in either group. (See Figure 7).

#### 3.3.4. Serotonin

Table 11 shows the results of analyzing changes in the level of salivary serotonin, which is a neurotransmitter related to happiness.

The results show that barefoot walking was more effective at changing serotonin levels than walking in sneakers, with a statistically significant difference between before and after the walking program, suggesting that walking barefoot in forests leads to a greater sense of emotional well-being. (See Figure 8).

## 4. Discussion

Immunity is the body’s defense mechanism that maintains homeostasis. Properly operating immune functions are essential to maintain homeostasis and prevent and treat various diseases [33]. Immune functions are affected by various factors, especially nutrition and motor ability. Although vigorous exercise and dietary restrictions diminish immune functions [34], proper exercise leads to increased innate immunity and stress resistance and reduced chronic inflammation, obesity, and insulin resistance [35]. Exercise intervention increases immunity in cancer patients [36], and regular exercise decreases cardiovascular disease, metabolic syndrome, diabetes, and inflammation [37,38,39]. Many studies have found a close correlation between inflammation and the immune system. Regarding inflammation, an imbalance between humoral and cellular immunity causes cancer [40]. To prevent this, the immune system’s response to inflammation causes phagocytosis or the neutralization of self-antigens [41]. Interleukin-6-mediated inflammation is related to diseased conditions such as insulin resistance, which is associated with diabetes and cancer [42]. Restoring the body’s immunity by normalizing the inflammatory response is crucial [43].

In general, CRP is an acute reactive substance generated in the liver and its level increases in the blood when acute inflammation or tissue damage occurs in the body. Its normal level is 61.29 ± 7.38 pg/mL. Therefore, it is used as a sensitive, rapid inflammation indicator to detect acute bacterial inflammation and inflammatory diseases. Increased CRP is associated with aging [44], obesity, diabetes, high blood pressure, and insulin resistance [45]. It is also related to physical inactivity [46]. Decreasing inflammation through physical activity favorably affects cardiovascular disease. Notably, exercise decreases CRP concentration by inhibiting insulin sensitivity and inflammation reactions [47]. This suggests that continuous aerobic exercise, such as forest walking, can reduce the CRP level.

In summary, we observe that blood circulation is one of the key factors in controlling inflammation and that stimulating the soles of the feet improves circulation and reduces inflammation. The experiment shows that CRP can vary greatly depending on the physical condition of the individual, and thus, it is necessary to check this condition in the initial experiment and to control it beforehand.

Second, IFN-γ, which is related to the immune system, is among the innate immune system’s main defenses, along with the T and B lymphocytes that constitute the lymphoid cells of NK and other immune cells. IFN-γ detects and removes viruses and pathogens that invade the body and finds and kills cancer cells generated by abnormalities in the host [48]. The 5.0–45.0 pg/mL range is generally considered normal. In both groups, IFN-γ continuously increased, starting at the beginning of the experiment, but there were no significant changes after the walking program’s midpoint. This suggests that the forest environment’s healing effect might have contributed to an initial increase. Indeed, the Korea Forest Service (2024) reported that forest activities independently improved NK cell activity. The above results also agree with those of previous studies [49,50], indicating significant changes in NK cells after five forest-walking exercise sessions over 3 days.

Third, serotonin, a neurotransmitter related to feelings of well-being, is secreted from the hypothalamic–pituitary axis in the brain. Excessive serotonin excites and stimulates brain function, whereas a lack of it causes depression by degrading brain function. Serotonin maintains balance and controls synaptic activity, which regulates body temperature, sensory perception, immune response, motor function, neuroendocrine control, appetite, learning, and memory [51]. Over 95% of serotonin is produced by the enteric nervous system, which is commonly referred to as the “second brain”, and only approximately 3% of serotonin is produced in the brain [52]. When a person is stressed, intestinal microorganisms secrete stress hormones that adversely affect the intestines, which may cause higher stress by inducing negative emotions through the limbic system [53]. Consequently, serotonin, a neurotransmitter that delivers feelings of well-being, decreases under stress, and dopamine is excessively produced, which may lead to aggression, depression, and suicidal impulses [54]. The 0.005–0.045 ng/mL range is considered normal for serotonin despite differences depending on age, sex, and health status. After the experiment, most of the barefoot group reported improved sleep quality, peace of mind owing to communion with nature, and a positive attitude in response to subjective questions about their feelings. The sneaker-wearing group reported improved physical strength, self-confidence, and vitality. These results were expected because barefoot forest walking relaxes the body and mind through contact with the ground, compared to walking in sneakers. Hence, barefoot walking leads to psychological stability through stress and tension relief and heightened physical stimuli and attention. In a previous study, the serotonin concentration increased in middle-aged women who walked and ran for 12 weeks [55]. The results of the barefoot walking experiment conducted in this study suggest that stimulating the soles of the feet, a sense of unity with nature, and the satisfaction derived from walking in a forest environment positively affected serotonin.

In summary, the results of this experiment show that in a rapidly changing society, external environmental factors can cause chronic and acute inflammation [56], leading to a collapse of the human immune system [57], which, in turn, leads to depression owing to a decrease in NK-cells, which act as anti-cancer agents, and a lack of serotonin.

The study makes a significant contribution to the literature by confirming the effects of barefoot forest walking on reduced inflammation, short-term changes in IFNγ, and positive effects on serotonin levels. The findings show that barefoot walking in urban forests leads to improved immunity, sleep quality, and emotional stability, owing to the combined effects of barefoot walking and the forest environment’s healing factors.

From this experiment, we learned that caution is required when navigating the forest terrain owing to uneven ground and that the nature of barefoot walking requires awareness of the risk of tetanus, knowledge of first aid, a proper walking form, and sufficient pre-stretching before starting the exercise. Barefoot walking can lead to accidents owing to the nature of the terrain, and thus, walking slowly and prioritizing emotional well-being through communing with nature should be a priority.

This study’s limitations and suggested future research directions are as follows. First, the study’s sample size may have been too small and localized to produce highly generalized results. Obtaining more generalizable results will require a sufficiently large nationwide sample. Importantly, interest in barefoot walking is growing, but evidence of its effects remains insufficient, necessitating further research. Second, given the proportional difference between male and female participants in this study, follow-up research should aim for a balanced sex division. Third, the physiological effects should be tracked after the participants have completed the experiment and returned to their daily lives to examine sustainability. Fourth, distinguishing grounding effects by comparing stationary barefoot standing with barefoot walking is necessary to verify the conduction effect during earthing. Fifth, future research should verify the effects of barefoot walking in different environments, such as beaches and school playgrounds, in addition to forests.

## 5. Conclusions

This study examined the physiological changes between walking in sneakers and walking barefoot in an urban forest environment. It found that barefoot walking decreased CRP as the experiment continued owing to the plantar stimulation and walking exercise effect; interferon gamma (IFNγ), a marker of NK cell activity, changed from the beginning of the experiment in both groups and did not change significantly after the middle of the experiment; and serotonin, known as the happiness hormone, was significantly higher in the barefoot group. Overall, the results suggest that barefoot walking in forests is more effective for improving health by improving immunity through healing factors through tactile, visual, auditory, and olfactory sensations through contact with the ground and by leading to emotional stability through a sense of connection with the body and nature. The results of this study can be used as a basis for further research on physiological changes to verify the effectiveness of barefoot walking in the future.

## Figures and Tables

**Figure 1 healthcare-12-02372-f001:**
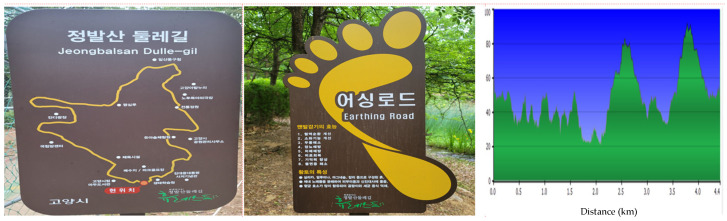
Barefoot walking course around Jeongbal Mountain.

**Figure 2 healthcare-12-02372-f002:**
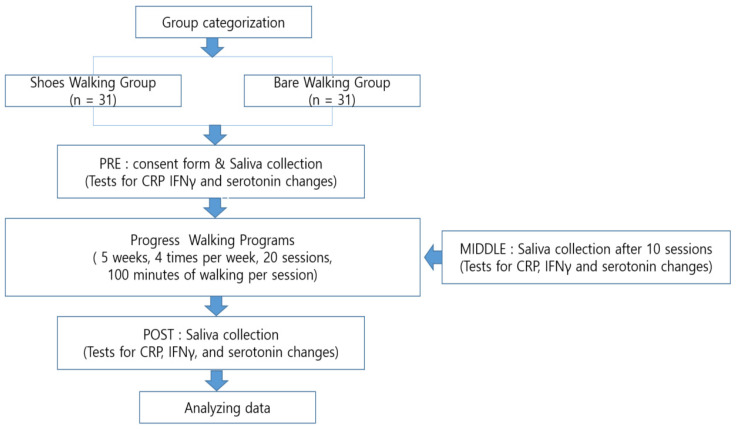
Experiment procedure.

**Figure 3 healthcare-12-02372-f003:**
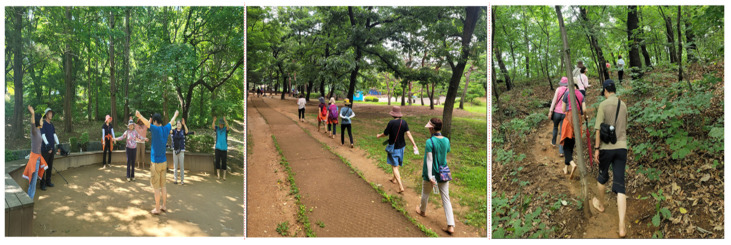
Participants walking barefoot around the circumference of Mt. Chung Bal.

**Figure 4 healthcare-12-02372-f004:**
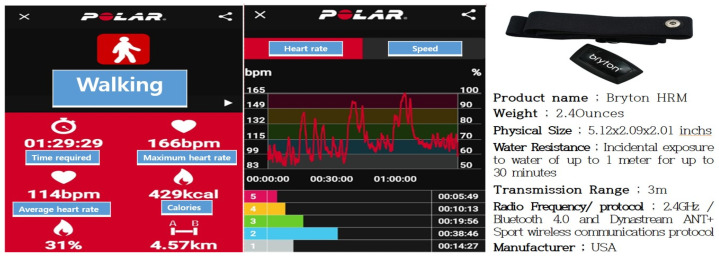
Measurement of walking exercise intensity.

**Figure 5 healthcare-12-02372-f005:**
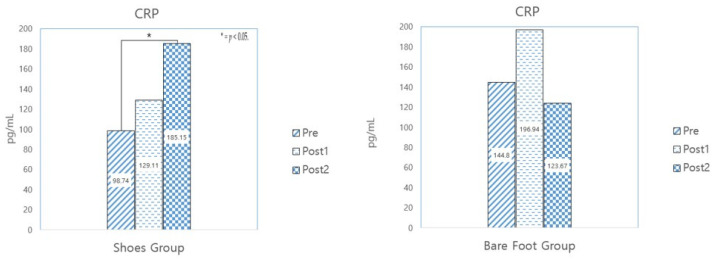
Barefoot and sneaker-wearing walkers’ paired sample *t*-test results for CRP.

**Figure 6 healthcare-12-02372-f006:**
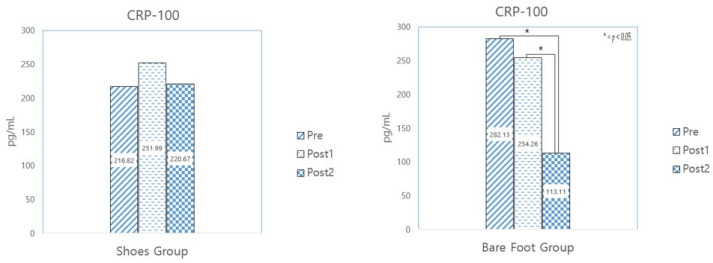
Results of a paired sample *t*-test analysis of CRP-100.

**Figure 7 healthcare-12-02372-f007:**
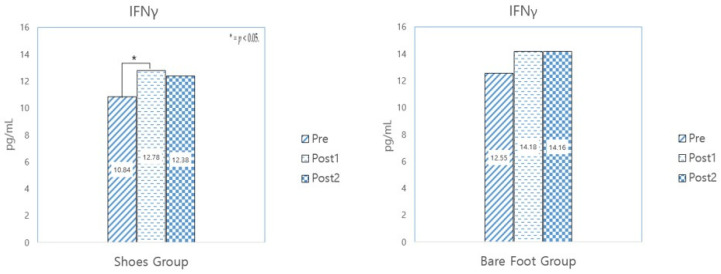
Results of a paired sample *t*-test analysis of IFNγ.

**Figure 8 healthcare-12-02372-f008:**
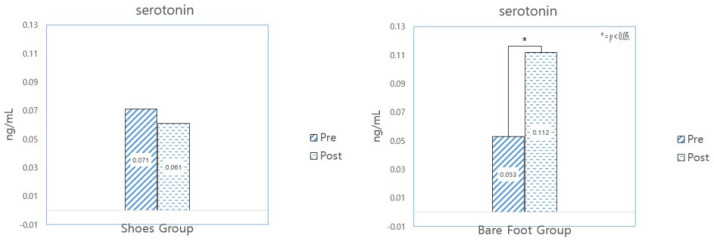
Results of a paired and independent sample *t*-test analysis of serotonin.

**Table 1 healthcare-12-02372-t001:** Participants’ age range by group.

Separation		All (*n* = 62)		SWG (*n* = 31)		BWG (*n* = 31)	
		*n*	%	*n*	%	*n*	%
Age range	40 s	5	8.1	4	12.9	1	3.2
	50 s	26	41.9	12	38.7	14	45.2
	60 s	27	43.5	12	38.7	15	48.4
	70 s	4	6.5	3	9.7	1	3.2

SWG = sneaker-wearing group; BWG = barefoot-walking group.

**Table 2 healthcare-12-02372-t002:** Walking program configuration.

Exercise	Duration	Frequency	Total Number	Distance	Workout Intensity	Activity Durations
Walking	5 weeks	4 times a week	20	4.4 km	THR; 50–70% RPE; 10–13	Warm-up: 5 min Main exercise: 90 min Cool-down: 5 min

THR = target heart rate; RPE = rate of perceived exertion.

**Table 3 healthcare-12-02372-t003:** Saliva analysis of substances and materials.

Substance Category	Analysis Material
CRP	Salivary CRP ELISA kit (Generation II), (Cat. No. 1-2102), Salimetrics Ltd. (Carlsbad, CA, USA)
IFNγ	Human IFNγ high-sensitivity ELISA kit, (ab236895), Abcam Ltd. (Cambridge, UK)
Serotonin	Serotonin ELISA kit, (ADI-900-175), Enzo Ltd. (Farmingdale, NY, USA)

CRP = C-reactive protein; IFNγ = interferon gamma.

**Table 4 healthcare-12-02372-t004:** Participants’ general characteristics.

Demographic	Variable	SWG (*N* = 31)	BWG (*N* = 31)	*p*
		M ± SE	M ± SE	
Age (years)	-	58.74 ± 4.46	59.58 ± 5.01	0.605
Age range	40 s	4 (12.9%)	1 (3.2%)	0.349
	50 s	12 (38.7%)	14 (45.2%)	
	60 s	12 (38.7%)	15 (48.4%)	
	70 s	3 (9.7%)	1 (3.2%)	
Sex	Male	6 (19.4%)	6 (19.4%)	1.00
	Female	25 (80.6%)	25 (80.6%)	
Alcohol consumption	Yes	7 (22.6%)	12 (38.7%)	0.168
	No	24 (77.4%)	19 (61.3%)	
Smoking	Yes	0 (0.0%)	1 (1.6%)	0.313
	No	31 (100%)	30 (98.4%)	
Regular exercise	Yes	24 (77.4%)	26 (83.9%)	0.520
	No	7 (22.6%)	5 (16.1%)	
Exercise routine (Number of weekly exercise sessions)	None	11 (35.5%)	8 (25.8%)	0.536
	Under 3	11 (35.5%)	10 (32.3%)	
	4 or more	9 (29.0%)	13 (41.9%)	
Daily exercise duration (h)	0	10 (32.3%)	8 (25.8%)	0.755
	Under 1	5 (16.1%)	3 (9.7%)	
	Under 2	10 (32.3%)	12 (38.7%)	
	2 or more	6 (19.3%)	8 (25.8%)	
Health condition	-	3.26 ± 0.86	3.35 ± 0.61	0.231
Regular eating habits	-	3.45 ± 0.85	3.42 ± 0.62	0.060
Favorite foods	Meat	0 (0.0%)	1 (3.2%)	0.455
	Vegetables	4 (12.9%)	6 (19.4%)	
	Balanced	27 (87.1%)	24 (77.4%)	

Values are mean ± standard deviation. SWG = sneaker-wearing group; BWG = barefoot-walking group.

**Table 5 healthcare-12-02372-t005:** The homogeneity test of physical characteristics.

Variable	SWG (*N* = 31)	BWG (*N* = 31)	*t*	*p*
Age (years)	58.74 ± 7.46	59.58 ± 5.01	−0.520	0.605
Height (cm)	160.84 ± 5.49	161.08 ± 0.35	−0.161	0.873
Weight (kg)	57.02 ± 7.40	59.55 ± 7.50	−1.338	0.186
BMI (kg/m^2^)	21.99 ± 2.14	22.90 ± 2.00	−1.720	0.091

Values are mean ± standard deviation. SWG = sneaker-wearing group; BWG = barefoot-walking group; BMI = body mass index.

**Table 6 healthcare-12-02372-t006:** The homogeneity test of physiological characteristics.

Variable	SWG (*N* = 31)	BWG (*N* = 31)	*t*	*p*
CRP (pg/mL)	98.75 ± 107.50	144.80 ± 191.29	−1.169	0.248
IFNγ (pg/mL)	10.84 ± 3.13	12.55 ± 4.25	−1.810	0.075
Serotonin (ng/mL)	0.070 ± 0.04	0.053 ± 0.04	1.669	0.100

Values are mean ± standard deviation. SWG = sneaker-wearing group; BWG = barefoot-walking group; CRP = C-reactive protein, inflammation indicator; IFNγ = interferon gamma, NK cell indicator; serotonin = happiness associator.

**Table 7 healthcare-12-02372-t007:** Normality tests.

Variable	Group	Skewness	Kurtosis	Shapiro–Wilk
Lg10_CRP (pg/mL)	SWG (*N* = 31)	−0.495	0.144	0.111
	BWG (*N* = 31)	−0.111	−1.138	0.153
Lg10_IFNγ (pg/mL)	SWG (*N* = 31)	−0.824	0.193	0.049
	BWG (*N* = 31)	−0.996	−0.027	0.002
Lg10_Serotonin (ng/mL)	SWG (*N* = 31)	−0.770	1.398	0.138
	BWG (*N* = 31)	−1.355	2.451	0.006

SWG = sneaker-wearing group; BWG = barefoot-walking group; CRP = C-reactive protein, inflammation indicator; IFNγ = interferon gamma, NK cell indicator; serotonin = happiness associator; Lg10(Commercial Logs); Skewness < 3; Kurtosis < 8, Shapiro–Wilk > 0.05.

**Table 8 healthcare-12-02372-t008:** Walking program’s effect on CRP.

Variable	Group	Pre	Post1	Post2	Pre-Post1		Post1-Post2		Pre-Post2	
		Before Program	After 10 Sessions	After 20 Sessions	*t*	*p*	*t*	*p*	*t*	*p*
CRP (pg/mL)	SWG (*N* = 31)	98.74 ± 107.50	129.11 ± 174.31	185.15 ± 213.76	−0.976	0.337	−1.265	0.216	−2.085	0.046 *
	BWG (*N* = 31)	144.80 ± 191.29	196.94 ± 232.65	123.67 ± 137.39	−1.110	0.276	1.156	0.140	0.519	0.608

* = *p* < 0.05. CRP = C-reactive protein, an inflammation indicator; SWG = sneaker-wearing group; BWG = barefoot-walking group.

**Table 9 healthcare-12-02372-t009:** Walking program’s effect on CRP-100.

Variable	Group	Pre	Post1	Post2	Pre-Post1		Post1-Post2		Pre-Post2	
		Before Program	After 10 Sessions	After 20 Sessions	*t*	*p*	*t*	*p*	*t*	*p*
CRP-100 (pg/mL)	SWG (*N* = 9)	216.82 ± 137.81	251.99 ± 265.98	220.67 ± 169.74	−0.345	0.739	0.308	0.766	−0.048	0.963
	BWG (*N* = 14)	282.13 ± 215.26	254.26 ± 233.00	113.11 ± 93.28	0.354	0.729	2.445	0.030	3.027	0.010 *

* = *p* < 0.05. CRP-100 = C-reactive protein-100 and over, an inflammation indicator; SWG = sneaker-wearing group; BWG = barefoot-walking group.

**Table 10 healthcare-12-02372-t010:** Walking program’s effect on IFNγ.

Variable	Group	Pre	Post1	Post2	Pre-Post1		Post1-Post2		Pre-Post2	
		Before Program	After 10 Sessions	After 20 Sessions	*t*	*p*	*t*	*p*	*t*	*p*
IFNγ (pg/mL)	SWG (*N* = 31)	10.84 ± 3.13	12.78 ± 2.47	12.38 ± 3.23	−3.095	0.004 *	0.555	0.583	−1.800	0.082
	BWG (*N* = 31)	12.55 ± 4.25	14.18 ± 4.29	14.16 ± 6.31	−1.624	0.115	0.017	0.986	−1.416	0.167

* = *p* < 0.05. IFNγ = interferon gamma, an NK cell indicator; SWG = sneaker-wearing group; BWG = barefoot-walking group.

**Table 11 healthcare-12-02372-t011:** Walking program’s effect on serotonin.

Variable	Group	Pre	Post	*t*	*p*
		Before Program	After 20 Sessions		
Serotonin (ng/mL)	SWG (*N* = 31)	0.071 ± 0.44	0.061 ± 0.62	0.710	0.483
	BWG (*N* = 31)	0.053 ± 0.040	0.112 ± 0.160	−2.081	0.046 *

* = *p* < 0.05. SWG = sneaker-wearing group; BWG = barefoot-walking group.

## Data Availability

Data are contained within the article.

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
