# Peer review of "Effects of Barefoot Walking in Urban Forests on CRP, IFNγ, and Serotonin Levels"

_healthcare, 2024, doi:10.3390/healthcare12232372_

Round 1
Reviewer 1 Report
Comments and Suggestions for Authors
This paper examines the health benefits of walking in the forest either wearing shoes or barefoot. Compared to previous studies on forest therapy, this study uses a relatively ample sample size and a longer observation period, which may offer extra evidence to reinforce previous findings.
Here are some personal suggestions for this study:
The background section of the introduction is somewhat lengthy and lacks sufficient literature support.
The introduction should strengthen the logical flow. Natural therapies related to forests are primarily used to address the disconnection between humans and nature in the context of global urbanization, as well as related health issues. This needs to be clearly stated in the introduction.
Over the past two decades, forest therapy, including forest walking, has received extensive attention. The authors should cite relevant reviews to introduce the overall research status, rather than listing many specific examples. Additionally, reviews on “green exercise” or “green physical activity” should also be mentioned, as many studies have been conducted under this theme and are closely related to forest therapy.
The authors want to emphasize the rationale for studying “barefoot walking,” but the research gaps in this area have not been well demonstrated. What has been overlooked in past forest walking studies? Why is it important? This needs to be addressed.
Barefoot walking itself has some health benefits; what additional benefits might there be when combined with natural exposure? Green exercise has been noted for its inclusion of natural exposure and physical activity simultaneously. The authors should consider and emphasize the necessity of this study from this perspective.
It is best to put up the research hypothesis at the end of the introduction, as this is not a data-driven exploratory study.
The authors should introduce their main focus indicators in the introduction, such as C-reactive protein. What physiological or psychological conditions do these indicators reflect?
Was the allocation of participants random? This needs more detailed explanation, as it could impact the sample size calculation and subsequent statistical methods.
All images are not visible in the text. Please make sure that it is not a systematic issue.
Please include a reference to the American College of Sports Medicine’s exercise guidelines.
Results should be presented and reported according to the CONSORT checklist. Please include a flowchart of the experiment and details of participant inclusion, exclusion, and final analysis process.
Since multiple measurements were taken from participants, why use the T-test? Statistical methods based on repeated measures should be used. For example, GEE.
The authors seem to only perform pre-post comparisons within a single group, which is confusing. Why not compare the two groups? This is the main evidence supporting the advantages of walking with barefoot over in sneakers. Differences within a single group at different time points should be a secondary or supplementary result.
The paper needs to be reorganized, as some texts in tables is unclear.
Additionally, the language needs revision. Some sentences are difficult to understand.
The discussion section is somewhat lengthy. Please condense it and focus on your findings.
Furthermore, barefoot walking poses some safety hazards, especially in forests. This should be mentioned in the discussion to enhance the practical application value of the research findings.
Comments on the Quality of English LanguageLanguage must be improved for clarity.
Author Response
Journal
Healthcare (ISSN 2227-9032)
Manuscript ID
healthcare-3185803
Author's Reply to the Review Report (Reviewer 1)
* Author's Notes to Reviewer
Here are some personal suggestions for this study:
The background section of the introduction is somewhat lengthy and lacks sufficient literature support.
We have slightly revised the background section of the Introduction and added additional references.
-> Please refer to the (1. Introduction) section of the attached revised manuscript (lines 52 to 66).
The introduction should strengthen the logical flow. Natural therapies related to forests are primarily used to address the disconnection between humans and nature in the context of global urbanization, as well as related health issues. This needs to be clearly stated in the introduction.
We have slightly revised the background section of the Introduction and added additional references.
-> Please refer to the (1. Introduction) section of the attached revised manuscript (lines 52 to 66).
Over the past two decades, forest therapy, including forest walking, has received extensive attention. The authors should cite relevant reviews to introduce the overall research status, rather than listing many specific examples. Additionally, reviews on “green exercise” or “green physical activity” should also be mentioned, as many studies have been conducted under this theme and are closely related to forest therapy.
We have slightly revised the background section of the Introduction and added additional references.
-> Please refer to the (1. Introduction) section of the attached revised manuscript (lines 52 to 66).
The authors want to emphasize the rationale for studying “barefoot walking,” but the research gaps in this area have not been well demonstrated. What has been overlooked in past forest walking studies? Why is it important? This needs to be addressed.
We have slightly revised the background section of the Introduction and added additional references.
-> Please refer to the (1. Introduction) section of the attached revised manuscript (lines 149 to 152).
Barefoot walking itself has some health benefits; what additional benefits might there be when combined with natural exposure? Green exercise has been noted for its inclusion of natural exposure and physical activity simultaneously. The authors should consider and emphasize the necessity of this study from this perspective.
We have slightly revised the background section of the Introduction and added additional references.
-> Please refer to the (1. Introduction) section of the attached revised manuscript (lines 152 to 155).
It is best to put up the research hypothesis at the end of the introduction, as this is not a data-driven exploratory study.
At the end of the introduction, we have added the research hypotheses of immune markers CRP, IFNγ, and serotonin.
- > Please refer to the end of the (1. Introduction) section of the attached revised manuscript (lines 171 ~173).
The authors should introduce their main focus indicators in the introduction, such as C-reactive protein. What physiological or psychological conditions do these indicators reflect?
The main focus markers of CRP, IFNγ, and serotonin, as well as physiologic and psychologic changes according to each saliva test, have been further described.
- > Please refer to section (1. Introduction) of the attached revised manuscript (lines 159~164).
Was the allocation of participants random? This needs more detailed explanation, as it could impact the sample size calculation and subsequent statistical methods.
A total of 62 participants were recruited and randomized into two groups after two stages stratifild sampling considering age, gender, etc.
- > Please refer to the (2. Materials and Methods) section of the attached revised manuscript (lines 182~185).
All images are not visible in the text. Please make sure that it is not a systematic issue.
We have added the corresponding images from the paper to be included in the text.
- > See Figures 1 through 8 for each corresponding
Please include a reference to the American College of Sports Medicine’s exercise guidelines.
Added relevant evidence to reference #61.
-> American College of Sports Medicine, ACSM's Guidelines for Exercise Testing and Prescription, www.acsm.org, 2013
Results should be presented and reported according to the CONSORT checklist. Please include a flowchart of the experiment and details of participant inclusion, exclusion, and final analysis process.
Revised manuscript (2. Materials and Methods) CONSORT checklist added and described.
- > To be revised for future manuscripts (Figures 3 and 4)
Since multiple measurements were taken from participants, why use the T-test? Statistical methods based on repeated measures should be used. For example, GEE.
In the case of CRP and IFNγ, Repeated Measures ANOVA analysis was performed to examine the interaction effect of the comparison between barefoot and sneaker groups, but the results were not statistically significant due to the large deviation of individual measurements, so we used paired sample T-test to intuitively express the mean change comparison and the difference value according to the time change, and tried to find out the statistical significance.
The authors seem to only perform pre-post comparisons within a single group, which is confusing. Why not compare the two groups? This is the main evidence supporting the advantages of walking with barefoot over in sneakers. Differences within a single group at different time points should be a secondary or supplementary result.
Although pre-homogeneity was achieved between the two groups, the independent sample t-test comparing the post-hoc means was not significant, so we used paired sample t-tests by group to express the change in mean over time and statistical significance.
The paper needs to be reorganized, as some texts in tables is unclear.
We have revised the manuscript to the attached revision.
Additionally, the language needs revision. Some sentences are difficult to understand.
We have revised the manuscript to the attached revision.
The discussion section is somewhat lengthy. Please condense it and focus on your findings.
We have revised the manuscript to the attached revision.
Furthermore, barefoot walking poses some safety hazards, especially in forests. This should be mentioned in the discussion to enhance the practical application value of the research findings.
In the discussion section, we have added the need to ensure the safety of barefoot walking.
- > Please refer to the (1. Discussion) section of the attached revised manuscript (lines 622-627).
Comments on the Quality of English Language
Language must be improved for clarity.
We have revised the manuscript to the attached revision.
Thank you for the informative review to help us with this experiment.
Submission Date
16 August 2024
Date of this review
28 Aug 2024 12:09:06
Reviewer 2 Report
Comments and Suggestions for Authors
The study aims to investigate the effects of barefoot walking on an urban forest path on C-reactive protein, interferon gamma, and serotonin levels. Although it is a well-designed and conducted study in general, I think that making some revisions I mentioned below will contribute to the development of the article.
The introduction section is too long and can be shortened.
In the materials and methods section, it is not stated how the distribution of the data is shown according to the arithmetic mean (Standard deviation, standard error). In fact, in order to decide to present the data with a mean or median, it is necessary to test the suitability of the data for a normal distribution. This process is also necessary for the selection of tests to be used in statistical analyses. The authors have preferred to use parametric tests to compare the arithmetic means of the groups, without performing a normality test before.
Arithmetic means are presented together with standard error in Table 4, while standard deviation is probably used in Table 7. Since the standard deviations are higher than the arithmetic averages, it can be assumed that the data presented in Table 7 have an abnormal distribution. Therefore, an independent sample t-test or a paired sample t-test should not be used. The Mann Whitney test or the Wilcoxon test should be used instead.
Figure 1, 2, 3, 4, 5, 6 and 7 doesn’t exist.
In lines 295-297, the authors said that a significant between-group difference in weight was observed: 57.02±7.40 kg for the sneaker-wearing group and 59.55±7.50 kg for the barefoot group (t = -0.161, p = 0.873). However, the p value (=0.873) they presented does not support this finding. Moreover, according to Table 5, this statistical analysis result belongs to height, not weight. The data presented in Table 5 are incompatible with the data presented between line 293 and 302. In addition, the data presented in tabular form does not need to be repeated as text. After the data in Table 5 have been revised by the authors, it will be appropriate to remove the text part.
Column headings are missing in Table 6. In the same way as Table 5, the data are presented in both text and tabular format in Table 6, 7, 8, 9 and 10. The data repeated should be removed.
The authors said that “Table 8 shows the results of analyzing CRP changes after selecting, from among all 62 participants, only those who exhibited a saliva CRP inflammation level of 100 pg/㎖ or higher before the experiment. The targets were 14 people from the barefoot group and nine from the sneaker-wearing group”. However, the number of participants is written as 31 for both two groups in Table 8.
In Table 10, the number of people whose serotonin levels were measured was indicated as nine in the SWG group and 14 in the BWG group. It should be explained in the methods section why serotonin levels of the other people in the groups were not measured.
The experimental procedure has been repeated many times in the method, results, and discussion sections. Unnecessary repetitions should be avoided.
Since the discussion section provides adequate and appropriate discussion in context, the unnecessary summary and repetition paragraphs between lines 563 and 585 should be removed.
The conclusion section should be revised. Because, the conclusion aims to help the reader understand why your research should be important to them after they have finished reading the article. A conclusion is a synthesis of key points, not just a summary of your points or a restatement of your research problem.
Author Response
Journal
Healthcare (ISSN 2227-9032)
Manuscript ID
healthcare-3185803
Type
Article
Title
Effects of Barefoot Walking in Urban Forests on CRP, IFNγ, and Serotonin Levels
Authors
Jaesun Kim * , Mimi Lee , Dongsoo Kim , Changseob Shin *
Author's Reply to the Review Report (Reviewer 2)
* Author's Notes to Reviewer
Comments and Suggestions for Authors
The study aims to investigate the effects of barefoot walking on an urban forest path on C-reactive protein, interferon gamma, and serotonin levels. Although it is a well-designed and conducted study in general, I think that making some revisions I mentioned below will contribute to the development of the article.
The introduction section is too long and can be shortened.
Only forest and barefooting related issues relevant to this study have been added.
-> Please refer to the introduction of the revised manuscript.
In the materials and methods section, it is not stated how the distribution of the data is shown according to the arithmetic mean (Standard deviation, standard error). In fact, in order to decide to present the data with a mean or median, it is necessary to test the suitability of the data for a normal distribution. This process is also necessary for the selection of tests to be used in statistical analyses. The authors have preferred to use parametric tests to compare the arithmetic means of the groups, without performing a normality test before.
In the case of C-reactive protein (CRP), which is a measure of inflammatory changes, it was expected that the level of inflammatory changes would be large in each subject. In order to conduct a parametric test rather than a non-parametric test, we calculated the appropriate number of people (n=62) for the T-TEST test through the G-POWER program. The purpose was to measure the level of change in the mean value of the calculation by group. We would greatly appreciate your consideration.
Arithmetic means are presented together with standard error in Table 4, while standard deviation is probably used in Table 7. Since the standard deviations are higher than the arithmetic averages, it can be assumed that the data presented in Table 7 have an abnormal distribution. Therefore, an independent sample t-test or a paired sample t-test should not be used. The Mann Whitney test or the Wilcoxon test should be used instead.
Table 4 shows the general characteristics of the subjects, and Table 7 is designed to show the mean value of CRP, which is the total annual change in CRP, by selecting the subjects (n=62) based on the t-test for parametric testing as described above.
Figure 1, 2, 3, 4, 5, 6 and 7 doesn’t exist.
We have inserted additional figures as shown in the revised manuscript.
In lines 295-297, the authors said that a significant between-group difference in weight was observed: 57.02±7.40 kg for the sneaker-wearing group and 59.55±7.50 kg for the barefoot group (t = -0.161, p = 0.873). However, the p value (=0.873) they presented does not support this finding. Moreover, according to Table 5, this statistical analysis result belongs to height, not weight. The data presented in Table 5 are incompatible with the data presented between line 293 and 302. In addition, the data presented in tabular form does not need to be repeated as text. After the data in Table 5 have been revised by the authors, it will be appropriate to remove the text part.
Table 5 has been modified to address your point, and only references the BMI index using age, height, and weight.
Column headings are missing in Table 6. In the same way as Table 5, the data are presented in both text and tabular format in Table 6, 7, 8, 9 and 10. The data repeated should be removed.
Partially revised column headings in Table 6 and briefly revised the physiological homogeneity text. We kept the text the same for additional physiologic explanations in Tables 6, 7, 8, 9, and 10.
The authors said that “Table 8 shows the results of analyzing CRP changes after selecting, from among all 62 participants, only those who exhibited a saliva CRP inflammation level of 100 pg/㎖ or higher before the experiment. The targets were 14 people from the barefoot group and nine from the sneaker-wearing group”. However, the number of participants is written as 31 for both two groups in Table 8.
Table 8 has been modified to show subjects in the sneaker group (n=9) and barefoot group (n=14).
In Table 10, the number of people whose serotonin levels were measured was indicated as nine in the SWG group and 14 in the BWG group. It should be explained in the methods section why serotonin levels of the other people in the groups were not measured.
The total number of subjects in Table 10 was 62, which was modified to sneaker group (n=31) and barefoot group (n=31).
The experimental procedure has been repeated many times in the method, results, and discussion sections. Unnecessary repetitions should be avoided.
Removed the experimental procedure section from the discussion section.
-> Delete lines 462-469
Since the discussion section provides adequate and appropriate discussion in context, the unnecessary summary and repetition paragraphs between lines 563 and 585 should be removed.
We've edited the entire discussion section for context.
The conclusion section should be revised. Because, the conclusion aims to help the reader understand why your research should be important to them after they have finished reading the article. A conclusion is a synthesis of key points, not just a summary of your points or a restatement of your research problem.
In the conclusion, I've added an explanation of why walking barefoot in the woods worked better.
Thank you for the informative review to help us with this experiment.
Submission Date
16 August 2024
Date of this review
03 Sep 2024 12:10:26
Reviewer 3 Report
Comments and Suggestions for Authors
Dear Authors,
Thank you for the opportunity to review your work, “Effects of Barefoot Walking in Urban Forests on CRP, IFNy, and Serotonin Levels.” This is an RCT study that involved 62 participants randomised into two groups (Sneakers vs. Barefoot walking) for 20 sessions of 90-minute walking over 5 weeks. This is a very interesting study. However, I have a few comments before the manuscript is considered for publication. I hope my suggestions can be useful in improving the manuscript for a wider reader population.
Comments:
1. Introduction: The introduction seems excessively long currently, even though it attempts to discuss many aspects of infectious diseases, cancer, and WHO’s definition of health. Many are irrelevant and non-specific to the study design and objectives. Too much verbiage may not improve audience engagement. I suggest streamlining the introduction; much of what’s covered on Page 2 can be omitted.
2. Methods: are the participants randomised into the 2 study groups or by participants’ preferences? This information is missing at this moment. If so, who performed the randomisation, and how was it done?
3. Methods: Were the walking sessions currently conducted? Are the sneaker and barefoot groups walking together during the sessions? What time were the walking sessions conducted? Morning, afternoon, or evening? Were there specific timings or randomised, too?
4. Information in Table 2 is described within the paragraphs. Suggest choosing either 1 format and reducing the length of the manuscript. It is rather long and filled with repeating information.
5. Since all figures are missing from the PDF document, I suggest re-uploading the submitted manuscript for further review.
6. Results: again, most of the descriptions in this section are self-explanatory via table presentations.
7. Would it be possible to combine Tables 4 and 5? They seem to follow a similar structure, except Table 5 has an additional column for reporting the t values.
8. Table 6 has no heading row for the analysis.
9. Discussion: Again, a lot of verbiage is used to describe simple concepts. Please consider reducing and only focusing on the key points.
Best Regards
Comments on the Quality of English LanguageThe manuscript is excessively long for the context and scale of this study.
Author Response
Journal
Healthcare (ISSN 2227-9032)
Manuscript ID
healthcare-3185803
Type
Article
Title
Effects of Barefoot Walking in Urban Forests on CRP, IFNγ, and Serotonin Levels
Authors
Jaesun Kim * , Mimi Lee , Dongsoo Kim , Changseob Shin *
Author's Reply to the Review Report (Reviewer 3)
* Author's Notes to Reviewer
Thank you for the opportunity to review your work, “Effects of Barefoot Walking in Urban Forests on CRP, IFNy, and Serotonin Levels.” This is an RCT study that involved 62 participants randomised into two groups (Sneakers vs. Barefoot walking) for 20 sessions of 90-minute walking over 5 weeks. This is a very interesting study. However, I have a few comments before the manuscript is considered for publication. I hope my suggestions can be useful in improving the manuscript for a wider reader population.
Comments:
- Introduction: The introduction seems excessively long currently, even though it attempts to discuss many aspects of infectious diseases, cancer, and WHO’s definition of health. Many are irrelevant and non-specific to the study design and objectives. Too much verbiage may not improve audience engagement. I suggest streamlining the introduction; much of what’s covered on Page 2 can be omitted.
In the revised paper, the literature related to barefoot walking and the need and purpose have been revised.
-> Please refer to the attached revised paper.
- Methods: are the participants randomised into the 2 study groups or by participants’ preferences? This information is missing at this moment. If so, who performed the randomisation, and how was it done?
A total of 62 participants were recruited and randomized into two groups after two stages stratifild sampling considering age, gender, etc.
- > Please refer to the (2. Materials and Methods) section of the attached revised manuscript (lines 182~185). In addition, we have added the consort checklist in Figures 2 and 3 to help you understand this study.
- Methods: Were the walking sessions currently conducted? Are the sneaker and barefoot groups walking together during the sessions? What time were the walking sessions conducted? Morning, afternoon, or evening? Were there specific timings or randomised, too?
The course was conducted in the same environment and at a fixed time, randomly and freely according to the subjects' choice, between 6:00 am and 8:00 pm, when they could go about their daily routine. The same time was not taken into account
- Information in Table 2 is described within the paragraphs. Suggest choosing either 1 format and reducing the length of the manuscript. It is rather long and filled with repeating information.
We have minimized redundancies throughout the revised manuscript.
- Since all figures are missing from the PDF document, I suggest re-uploading the submitted manuscript for further review.
We have included Figures 1 through 10 in the revised manuscript.
- Results: again, most of the descriptions in this section are self-explanatory via table presentations.
Textual additions to Tables 1-10 have been made to minimize the description.
- > Please refer to the full revised paper.
- Would it be possible to combine Tables 4 and 5? They seem to follow a similar structure, except Table 5 has an additional column for reporting the t values.
We hope you understand that Table 5 was added separately to validate BMI homogeneity for the experimenter's body.
- Table 6 has no heading row for the analysis.
Added heading row in Table 6.
-> Please refer to line 343 of the revised paper.
- Discussion: Again, a lot of verbiage is used to describe simple concepts. Please consider reducing and only focusing on the key points.
Overall, we've tried to make the revised manuscript the best it can be.
Thank you for the informative review to help us with this experiment.
Best Regards
Comments on the Quality of English Language
The manuscript is excessively long for the context and scale of this study.
Submission Date
16 August 2024
Date of this review
05 Sep 2024 07:35:58
Round 2
Reviewer 2 Report
Comments and Suggestions for Authors
The introduction section has been made even longer.
The Methods section does not specify how the data distribution is presented (SD or SE), the normal distribution compliance test was not performed, and my suggestions for choosing a statistical test were not considered.
As I mentioned in my previous evaluation in Tables 7, 8 and 9, the findings continue to be repeated both in the table and in the text.
Variable name in the first column disappeared in Table 9, the table title was located at the bottom of the table.
I said in a previous review that “A total of 26 (43%) of the sources contain research older than ten years.” However, the authors did not make any changes, but added another reference that was not used in the manuscript.
Reviewer 3 Report
Comments and Suggestions for Authors
Dear Authors,
Please remove Figure 4 (Consort Checklist) from the manuscript. The checklist should only be included as a supplementary file instead of appearing in the main manuscript.
Comments on the Quality of English Language
nil
Round 3
Reviewer 2 Report
Comments and Suggestions for Authors
I am glad to see the changes and corrections made by the authors. I thank them for addressing the points raised. While they may not have updated the literature references sufficiently, if the editorial team deems it acceptable, there is no issue.
Author Response
Comment1
Comments and Suggestions for Authors
I am glad to see the changes and corrections made by the authors. I thank them for addressing the points raised. While they may not have updated the literature references sufficiently, if the editorial team deems it acceptable, there is no issue.
<<Response>>
- To improve the quality of the manuscript according to the editorial team's requirements, we sent the revised manuscript to a professional translation agency (Editage) for proofreading. We have sent the finalized proofread manuscript as an attachment below.
In the meantime, thank you for your time in revising the manuscript.

Reviewer 3 Report
Comments and Suggestions for Authors
Dear Authors,
Thanks for the revision. I notice you have made major revisions to the manuscript. However, I suggest some major rewriting based on the current English standard. The current writing approach may not have met the standard for manuscript publication.
Comments on the Quality of English LanguageA major revision is necessary.
Author Response
<<Comments>>
Dear Authors,
Thanks for the revision. I notice you have made major revisions to the manuscript. However, I suggest some major rewriting based on the current English standard. The current writing approach may not have met the standard for manuscript publication.
Comments on the Quality of English Language
A major revision is necessary.
<<Response>>
- To improve the quality of the manuscript according to the editorial team's requirements, we sent the revised manuscript to a professional translation agency (Editage) for proofreading. We have sent the finalized proofread manuscript as an attachment below.
In the meantime, thank you for your time in revising the manuscript.

Round 4
Reviewer 3 Report
Comments and Suggestions for Authors
Dear Authors,
Thank you for the revision. I appreciate your relentless efforts with the paper revision. With the latest revision, with the help of a professional English editor, the manuscript seems to be written in a more conventionally accepted English format.
I have no further comments.
Comments on the Quality of English Language
Dear Authors,
It may be easier for the reviewer if the current draft is presented in a less marked-up style.